# Targeting the P10 Peptide in Maturing Dendritic Cells via the DEC205 Receptor In Vivo: A New Therapeutic Strategy against Paracoccidioidomycosis

**DOI:** 10.3390/jof9050548

**Published:** 2023-05-10

**Authors:** Suelen S. Santos, Eline Rampazo, Carlos P. Taborda, Joshua D. Nosanchuk, Silvia B. Boscardin, Sandro R. Almeida

**Affiliations:** 1Department of Clinical and Toxicological Analysis, Faculty of Pharmaceutical Sciences, University of São Paulo, São Paulo 05508-000, SP, Brazil; 2Department of Parasitology, Biomedical Sciences Institute, University of São Paulo, São Paulo 05508-000, SP, Brazil; 3Department of Microbiology, Biomedical Sciences Institute, University of São Paulo, São Paulo 05508-000, SP, Brazil; 4Departments of Medicine, Division of Infectious Diseases, Microbiology and Immunology, Albert Einstein College of Medicine and Montefiore Medical Center, Bronx, NY 10461, USA

**Keywords:** *Paracoccidioides brasiliensis*, paracoccidioidomycosis, dendritic cells, DEC205, P10 peptide

## Abstract

Paracoccidioidomycosis (PCM) is a systemic mycosis caused by *Paracoccidioides brasiliensis*, a thermally dimorphic fungus, which is the most frequent endemic systemic mycosis in many Latin American countries, where ~10 million people are believed to be infected. In Brazil, it is ranked as the tenth most common cause of death among chronic infectious diseases. Hence, vaccines are in development to combat this insidious pathogen. It is likely that effective vaccines will need to elicit strong T cell-mediated immune responses composed of IFNγ secreting CD4^+^ helper and CD8^+^ cytolytic T lymphocytes. To induce such responses, it would be valuable to harness the dendritic cell (DC) system of antigen-presenting cells. To assess the potential of targeting P10, which is a peptide derived from gp43 secreted by the fungus, directly to DCs, we cloned the P10 sequence in fusion with a monoclonal antibody to the DEC205 receptor, an endocytic receptor that is abundant on DCs in lymphoid tissues. We verified that a single injection of the αDEC/P10 antibody caused DCs to produce a large amount of IFNγ. Administration of the chimeric antibody to mice resulted in a significant increase in the levels of IFN-γ and IL-4 in lung tissue relative to control animals. In therapeutic assays, mice pretreated with αDEC/P10 had significantly lower fungal burdens compared to control infected mice, and the architecture of the pulmonary tissues of αDEC/P10 chimera-treated mice was largely normal. Altogether, the results obtained so far indicate that targeting P10 through a αDEC/P10 chimeric antibody in the presence of polyriboinosinic: polyribocytidylic acid is a promising strategy in vaccination and therapeutic protocols to combat PCM.

## 1. Introduction

Dendritic cells (DC) are antigen-presenting cells that act as sentinels in peripheral tissues, constantly sampling the antigens in their environment [1]. The DC population lining the lungs plays a key role in the initiation of T cell responses after pulmonary challenge with certain microbes [2]. In several models of infectious diseases, DCs are being studied for their ability to serve as a vaccine adjuvant and mediating protection against bacteria, viruses, parasites, or fungal pathogens [3]. In the last decade, researchers have attempted to study the targeting of antigens to DCs by using monoclonal antibodies (mAb) against DC receptors fused to antigens of interest [4,5,6,7]. One of the two major resident DC populations in mouse secondary lymphoid organs expresses the CD8α chain and a C-type lectin endocytic receptor known as DEC205/CD205 (CD8α^+^DEC205^+^). Some works have demonstrated that animals vaccinated with mAb αDEC205 induce large numbers of Th1 type CD4^+^ T, as well as CD4^+^ T cell-dependent protection to microbial challenge [7,8,9].

In Brazil and other Latin America countries, paracoccidioidomycosis (PCM) is the most prevalent deep mycosis [10]. PCM is caused by the thermally dimorphic fungi *Paracoccidioides brasiliensis* and *Paracoccidioides lutzii,* as well as the more recently described *Paracoccidioides americana, Paracoccidioides restrepiensis*, and *Paracoccidioides venezuelensis* [11].

There are around 10 million people infected, of whom it is estimated that 2% will develop clinical disease [12]. In addition, in high endemicity areas such as Brazil, the annual incidence is 3 cases per 100,000 inhabitants, with a fatality rate between 2 and 23% [13]. The primary treatment of PCM is the use of antifungal drugs, but nutritional support, the management of sequelae, and treatment of additional comorbidities are also important for the successful resolution of PCM. Moreover, antifungal chemotherapy is administered for a protracted duration (usually over than 2 years), which leads to a growing concern regarding drug toxicity, treatment costs, and high dropout rates. However, this prolonged approach to PCM does not ensure the complete destruction of the fungus [14]. In this context, the development of a vaccine or effective immunotherapy could be used in endemic areas to prevent new cases of the disease and/or modify the duration of treatment.

Puccia et al. (1991) [15] described gp43, 43 kDa glycoprotein of *P. brasiliensis*, which is recognized by 100% of sera from patients with PCM but not by sera from controls or individuals with other systemic mycoses. Based on the sequence of gp43, which encodes a polypeptide of 416 amino acids, Taborda et al. (1998) [16] found that only a peptide consisting of 15 amino acids (QTLIAIHTLAIRYAN; named P10) induced lymphocyte proliferation in cells from mice previously immunized with the full glycopeptide gp43 or the P10 fragment. P10 effectively induces the production of CD4^+^ T cells, and its potential use in vaccines has been demonstrated in several studies [17,18,19,20,21,22].

In the present work, we cloned the P10 sequence in fusion with a monoclonal antibody (mAb) to the DEC205, an important receptor related to endocytosis on DCs in lymphoid tissues [23]. Already knowing that an antigen targeting DEC205 in the absence of a DC maturation stimulus leads to tolerance, we used polyriboinosinic: polyribocytidylic acid (poly (I:C)) as adjuvant. This molecule is a synthetic double-stranded RNA previously described to induce a Th1 type of response against viruses, bacterium, and protozoan parasites through the activation of innate immunity via toll-like receptor (TLR) 3 and melanoma-differentiation-associated gene-5 (MDA5), endosomal and cytoplasmic receptors, respectively [7,24,25]. Our results show that a single immunization of mice with the αDEC/P10 mAb induced high amounts of IFN-γ. In therapeutic assays, animals treated with αDEC/P10 presented a decrease in fungal burden and displayed minimal pulmonary damage. Hence, the αDEC/P10 chimera holds significant promise as a therapeutic agent.

## 2. Methods

### 2.1. Animals and Ethics Statement

The murine studies were approved by the Committee on the Ethics of Animal Experiments of the University of Sao Paulo (Permit Number: CEUA 292). BALB/c male mice (8–12 weeks old) were obtained from the specific pathogen-free facility of the University of Sao Paulo and maintained in our animal facility in a manner conforming to institutional guidelines for animal care and welfare.

### 2.2. P. brasiliensis Strains and Mouse Infection

The yeast form of the highly virulent *P. brasiliensis* strain 18 was grown in Sabouraud-agar. The strain that was used was recovered from animals before the experiments. A suspension of fungi was prepared with sterile PBS (137 mM NaCl, 2.7 mM KCl, 10 mM Na_2_HPO_4_, 2 mM KH_2_PO_4_), and yeast cells were adjusted to 1 × 10^6^ cells in 50 μL, based on hemocytometer counts. Mice were anesthetized and challenged with an intratracheal inoculation of 1 × 10^6^ Pb strain 18 yeast cells.

### 2.3. Plasmid Generation

The sequence encoding the P10 sequence was cloned in frame with the carboxyl terminus of the heavy chain of mouse aDEC205 (NLDC145 clone) or an isotype control antibody (kindly provided by Dr. Silvia Beatriz Boscardin, University of São Paulo), as previously described [4,6,7,26]. Amplification was accomplished by using the Phusion High Fidelity DNA Polymerase (New England Biolabs, Ipswich, MA, USA) according to the manufacturer’s instructions. Plasmids pDEC205-empty, pDEC205-P10, and pISO-P10 were then generated and sequenced to confirm the presence of P10 in frame.

### 2.4. Expression of Recombinant Fusion Antibodies

DH5α bacteria were transformed by using a plasmid containing the heavy chain of the mouse aDEC205 or isotype control (pDEC205-empty, pDEC205-P10, and pISO-P10) and their respective light chain (pDEC205 kappa, kindly provided by Dr. Silvia Boscardin, University of São Paulo) as previously described [27].

Briefly, human embryonic kidney (HEK) 293T (ATCC No CRL-11268) was transfected with polyethyleneimine (PEI) and 10 mg of each sample of vector DNA purified with QIAGEN Maxi Prep columns (Qiagen, Hilden, Germany) corresponding to heavy and light chain [28], in a final volume of 1 mL of 150 mM NaCl solution containing 4.5 mg of PEI per mg of DNA. The mix was distributed evenly on the culture plate. Culture supernatants were harvested 5–6 days after transfection and cleared from cell debris via centrifugation, and the antibodies present in the supernatants were precipitated by the addition of ammonium sulfate (Amresco, Boise, ID, USA). The precipitated antibodies were suspended in 50 mL of cold PBS containing 1 mM PMSF (Amresco) and dialyzed against 2 L of cold PBS. Recombinant fusion antibodies were purified with Protein G beads (GE Healthcare, Chicago, IL, USA). The fractions containing antibodies were pooled together and dialyzed against 2 L cold PBS and filtered through 0.2 mM membranes (TPP) for sterilization, and finally, their concentrations were estimated via the Bradford assay (Pierce, Los Angeles, CA, USA). Aliquots were then stored at 20 °C until use.

### 2.5. Antibody Viability Assay

Spleens from C57BL/6 mice were collected and macerated, and the cells were separated by using a cell strainer (BD Falcon 100 micron). The cell suspensions were kept on ice. After the first wash, the erythrocytes were removed by using a lysis solution. The splenocyte cell solution was suspended, and the number of cells were estimated by counting in a Neubauer chamber, using Trypan Blue solution to observe their viability. Cells were incubated with antibodies (anti-CD16/32) that recognize and block the Fc receptors on the membrane of cells at a concentration of 1:100. After 15 min, cells were plated 5 × 10^6^ cells/well and incubated with 3 different concentrations of the hybrid antibodies (10 µg/mL, 1 µg/mL, and 0.1 µg/mL). After 40 min, cells were washed with FACS buffer and incubated with αIgG1 conjugated to Phycoerythrin (PE). All mAbs were purchased from BD biosciences. After another 40 min, the cells were washed and subjected to flow cytometry. The results were analyzed by using FlowJo software.

### 2.6. Immunizations

Mice were challenged with a single intraperitoneal (i.p.) dose of 5 µg of mAb αDEC/P10 or ISO/P10, or 20 µg of the peptide P10 without any mAb. The mAbs or P10 were administered in the presence of 50 µg of poly (I:C) diluted in sterile PBS. An additional group received poly(I:C) only.

### 2.7. Analysis of T CD4^+^ IFN-γ Producing Cells in Animals Immunized with Hybrid Antibodies (ELISPOT)

For the ELISPOT assay, we used the “Ready-Set-Go” (eBioscience, San Diego, CA, USA) kit for the detection of IFN-γ and nitrocellulose plates with 96 wells (MAIPS 4510, Millipore, Burlington, MA, USA) according to the manufacturer’s instructions. The day before splenocyte isolation, the capture antibody was diluted in sterile buffer that was specific for sensitization of the PVDF membrane of the ELISPOT plate. After the addition of 100 mL per well of diluted antibody, the plate was incubated at 4 °C for 16–18 h and then blocked by using RPMI supplemented with 10% sterile FBS for 1 h at room temperature. Splenocytes were plated at 3 × 10^5^ cells per well, and medium containing rIL-2 at a final concentration of 30 U/mL and peptide P10 at a concentration of 20 mg/mL were added to their respective wells. A medium containing rIL-2 and 2.5 mg/mL Concanavalin A was used as a positive control. As a negative control, only rIL-2 was used, and the cells were stimulated in vitro with an unrelated peptide (Tewet). Cells were incubated in a CO_2_ incubator at 37 °C for 24 h. To remove any residual cells, the plates were washed 3 times with PBS containing 0.05% Tween 20. A detection antibody was added for 2 h at room temperature, the plate was washed again, and streptavidin HRP × 250 was added. After 45 min, the plate was washed and developed by using AEC substrate (Becton Dickinson, Franklin Lakes, NJ, USA) for approximately 30 min. The plates were then dried at room temperature, and spots were counted in an automated counter (AID GmbH, Strassberg, Germany).

### 2.8. Therapy Protocol

We evaluated the chimeric antibody as a therapeutic vaccine for PCM. Mice were infected as described and monitored for 45 days. The infected mice then intraperitoneally (i.p.) received either 5 µg of antibody αDEC/P10, the antibody anti-DEC205 not fused (αDECempty), or 20 µg of the peptide P10 without any antibody. The mAbs and P10 were co-administered with 50 µg of poly (I:C) diluted in sterile PBS. A fourth group received poly (I:C) alone. The administration of either mAb or P10 was in 3 doses with an interval of 7 days between them. After 10 days of the end of the treatment (10 days after the third dose), the animals were euthanized, and lungs, livers, and spleens were evaluated.

### 2.9. Cytokines Measurements

IFN-γ, IL-10, and IL-4 cytokine production was assayed in the supernatant of homogenate of lungs, livers, and spleens. The measurement was performed by using commercial kits in accordance with the R&D manufacturer’s instructions.

### 2.10. Colony-Forming Unit (CFU) and Histopathology Assays

The tissues were removed and weighed. They were macerated, and aliquots were plated onto BHI agar supplemented with equine serum and 5% culture filtrate of the isolate 192 of *P. brasiliensis* as a growth factor, in addition to the antibiotic penicillin (0.062 mg/mL) and streptomycin (100 μg/mL). After 2 weeks of incubation at 37 °C, the resulting colonies were enumerated to obtain the number of colonies per gram of organ.

For the preparation of histopathological slides, tissues were placed in fixative solutions, embedded in paraffin, sectioned, and stained with hematoxylin and eosin or silver nitrate (Gomori). Histological sections were analyzed and photographed under an optical microscope (200× and 400×).

## 3. Results

### 3.1. Hybrid Antibody αDEC/P10 Binds Specifically to a DEC205^+^ DC

The ability of the αDEC/P10 chimera to bind specifically to DEC205^+^ DCs was evaluated. Using three different concentrations of the fused antibody, we verified a dose-dependent binding of the mAb via flow cytometry, confirming the specific αDEC/P10 mAb for CD19^−^DX5^−^MHCII^+^CD11c^+^CD8^+^ cells, a subpopulation of DCs that express the receptor DEC205 (Figure 1).

### 3.2. Evaluation of the Specific Response by IFN-γ-Producing Cells after Immunization with the Chimeric Antibody αDEC/P10

In order to analyze whether the constructed αDEC/P10 chimera could activate IFN-γ-producing cells in vivo, the animals were immunized and restimulated in vitro. Hence, we first focused our attention on IFN-γ T CD4^+^ producing cells, as the P10 is an MHC-II binding peptide. The production of IFN-γ is protective in PCM [29]. We evaluated the cellular response in splenocyte cultures 10 days after immunization with the different test constructs, stimulated with peptide P10 (20 ug/mL) or with an unrelated control peptide named TEWETGQI. We found that a single dose of the chimeric αDEC/P10 mAb resulted in a specific CD4^+^ T response, which was significantly higher than immunization with peptide alone and other control groups (Figure 2).

### 3.3. Cytokine Production in Organs after Treatment by Using Chimeric Antibodies in Animals Infected with P. brasiliensis

We evaluated the production of IFN-γ, IL-4, and IL-10 in the livers, spleens, and lungs of animals 10 days after the 3rd immunization. The lungs are the major organ involved in PCM, and our results demonstrate that the administration of αDEC/P10 chimera results in a significant decrease in the amount of IFN-γ produced in this organ in relation to the control, but there was no difference with the P10 group (Figure 3A), although there was no difference detected in relation to IL-4 (Figure 3B). When we verified the IL-10 production, we verified that the group that received the αDEC/P10 produces a lower amount of this cytokine in comparison with all groups analyzed (Figure 3C).

In the spleen, the production of IFN-γ and IL-4 was similar between the different treatment groups, but IL-10 was lower in the group treated with the αDEC/P10 chimera in relation to P10 (Figure 3F). In the liver, higher levels of IFN-γ and IL-4 were measured in the group treated with αDEC/P10 mAb compared to the other groups (Figure 3G,H).

### 3.4. Fungal Load Reduction in ANIMALS Treated with Chimeric αDEC/P10 Antibody

To examine the efficacy of the αDEC/P10 chimera, we evaluated the pulmonary CFUs and lung architecture from mice 10 days after the completion of the third administration of the chimera and compared these findings to the other groups. Notably, there was a significantly lower fungal burden in the animals treated with αDEC/P10 mAb (Figure 4). Similarly, the histopathological evaluation of lung tissues from mice treated with αDEC/P10 mAb revealed an absence of yeast cells, and the lung architecture appeared normal (Figure 5A). In contrast, control mice that received PBS alone displayed the greatest number of fungal cells, and an intense inflammatory infiltrate was present (Figure 5F). It was also possible to note that other treatments were not effective in controlling pulmonary fungal infection (Figure 5B–E). These results demonstrate the efficiency of the αDEC/P10 chimera in the control of *P. brasiliensis* lung infection.

## 4. Discussion

Chemotherapy is the basis of treatment of PCM in its various forms; however, it can be extended for long periods with an alarming frequency of relapses. There is a consensus that the therapeutic vaccine instead of the prophylactic one would be the best way to control PCM. Several works have demonstrated that a therapeutic vaccine to boost the cellular immune response seems relevant not only to reduce the time of treatment but also to prevent relapses and improve the prognosis of anergic cases [30,31,32]. P10 has previously been well characterized as an epitope for CD4^+^ T cells [16], and many studies have shown the ability of this peptide to generate a protective immune response against *P. brasiliensis* in different preparations (Freund’s complete adjuvant, Salmonella enterica flagellin FliC) [16,20]. To further utilize the promise of P10 in a vaccine strategy, we created the αDEC/P10 chimera and demonstrated that it binds specifically and dose-dependently to DCs.

Protection against systemic fungal infection requires an effective cell immune response, and the presence of IFN-γ appears to be essential to control the progression of PCM [16,29,33]. We started immunizing the animals with 5 μg of αDEC/P10 and 50 μg of poly (I:C) and found that the targeting was able to stimulate a CD4^+^ T cell response as shown by the ELISPOT, by which we detected significant numbers of IFN-γ-producing cells. This response was significantly higher than what we found in animals that received 20 μg of P10, also in the presence of poly (I:C). It is also important to note that the dose of P10 that was used was 4-fold greater than that of the chimeric antibody, resulting in a dose of P10 administered that was about 200-fold greater relative to the concentration of P10 in the chimera.

The PCM can present as a severe and disseminated form involving the lungs, skin, lymph nodes, spleen, liver, and lymphoid organs of the gastrointestinal tract [34]. In order to analyze the immune response in the different organs such as lung, spleen, and liver, the cytokine production was quantified. By using the chimeric antibody to treat PCM, we found that after treatment, the group treated with αDEC/P10 had almost basal production of IFN-γ and IL-10, and all groups produced large amounts of IL-4. Arruda et al. (2004) [35] showed that IL-4 may play different roles in pulmonary PCM, depending on the genetic pattern of the host. Depletion of IL-4 in B10.A (genetically susceptible mice) led to increased lung fungal load, while the in vivo depletion of endogenous IL-4 in a PCM model at C57BL/6 (intermediate sensitivity to *P. brasiliensis*) was less severe and was associated with increased production of TNF-α and IL-12 and with decreased secretion of IL-4 and IL-5. These results are different from those shown for other systemic mycoses, such as candidiasis, histoplasmosis, and coccidioidomycosis, in which IL-4 depletion in vivo induces immunoprotein and a polarization for Th1 responses greater than Th2. One explanation is that IL-4 is not the main mediator that governs PCM susceptibility in B10.A mice. The antagonistic effects of IL-4 on genetically different hosts demonstrate that several immunological mechanisms may lead to susceptibility to *P. brasiliensis* lung infection, and more importantly, the Th1-Th2 paradigm does not explain all the immunological mechanisms that determine the evolution of disease [35]. In general, DCs stimulated with IFN-γ cause mature cells to preferentially produce IL-12, whereas DCs stimulated by IL-10 and prostaglandin E2 generate low amounts of IL-12 [36]. We believe that this low amount of IL-10 and IFN-γ is because the disease is controlled by the treatment. Assessing the levels of these cytokines at earlier times during the infection and treatment may reveal additional differences into the biological effects of the αDEC/P10 chimera.

The αDEC/P10 chimera showed significant protective efficacy as measured by the major reduction in the pulmonary fungal load and the resolution of lung disease shown in histopathological evaluation. The results obtained by De Amorin et al. (2013) [22], who used a DNA vaccine (pcDNA3-P10), Th1-related cytokines remained high during treatment, in addition to the increase in IL-10, and with this cytokine profile, although distinct from those found in our study, the mice that received the pcDNA3-P10 largely resolved their infections. It is believed, in this case, that the regulatory cells were responsible for the normal histopathology of the treated tissue. Notably, Rittner et al. (2012) [21] demonstrated that the tissue architecture normalized only after the administration of 7 doses of P10 and IL-12 DNA over 6 months of treatment. Another recent study demonstrated that lung homogenates from immunosuppressed and Pb18-infected mice treated with DCs pulsed with P10, with or without antifungal drugs, contained significantly higher levels of IL-12, and there was a significant reduction in IL-10 levels compared with the lungs of immunosuppressed and infected mice that did not receive DCs [37]. It is important to note that at the beginning of the infection, the Th1 pattern is important in the response to the fungus, but for prolonged periods, exacerbated inflammation can damage the tissue, and this is often the cause related to the aggravation of the disease, with fibrosis and loss of tissue function.

P10 is a major immunoprotective antigen that is being harnessed in vaccine development, and so far, it is the only candidate for vaccination in PCM. The administration of P10 either before or after the establishment of PCM produces a therapeutically protective effect in both immunosuppressed and immunocompetent mice [32]. P10 requires concomitant administration of adjuvant for efficacy, although diverse adjuvants produce a therapeutic effect with the peptide [31], including delivery as a P10-nanoparticle [30].

We have demonstrated that the targeting of P10 through its delivery through and display in a chimeric αDEC/P10 antibody in the presence of poly (I:C) is a promising strategy for vaccination against *P. brasiliensis,* and this strategy has been demonstrated to be a more efficient protocol when compared with only peptide immunization. The immunomodulatory effects of this approach appear to facilitate the clearance of the fungus while concomitantly reducing host damage, as demonstrated by the normalization of lung architecture after pulmonary challenge. Given that αDEC205 antibody has been shown to be safe and effective in human studies [38], our results indicate that further investigations with our αDEC/P10 chimera are warranted to define their efficacy as a prophylactic and therapeutic modality against *P. brasiliensis*.

## Figures and Tables

**Figure 1 jof-09-00548-f001:**
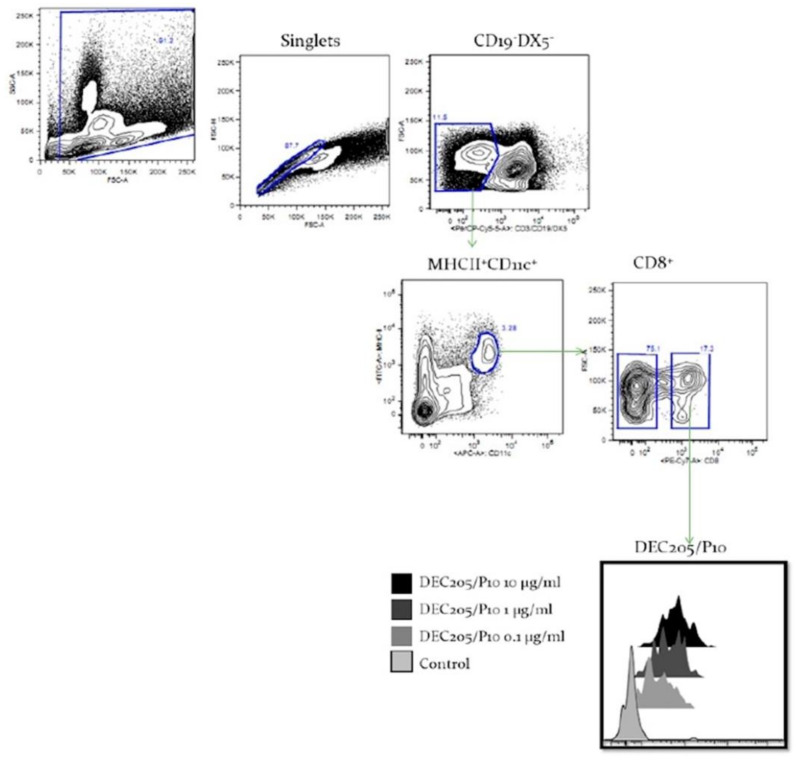
The αDEC205/P10 chimeric antibody binds to CD8α^+^ dendritic cells. Total splenocytes from naïve C57BL/6 mice were incubated with αDEC/P10 antibodies at 3 concentrations (10 µg/mL, 1 µg/mL, and 0.1 µg/mL). After incubation, the splenocytes were labeled with antibodies to CD19, MHCII, DX5, CD11c, and CD8. DCs were gated as CD19^−^ DX5^−^MHCII^+^CD11c^+^, as subdivided in CD8α^+^ and CD8α^−^. The binding of different concentrations of the αDEC/P10 to the CD8α^+^ DCs is shown in the histograms by using an anti-mouse IgG1 as a secondary antibody. As control was used only an anti-IgG.

**Figure 2 jof-09-00548-f002:**
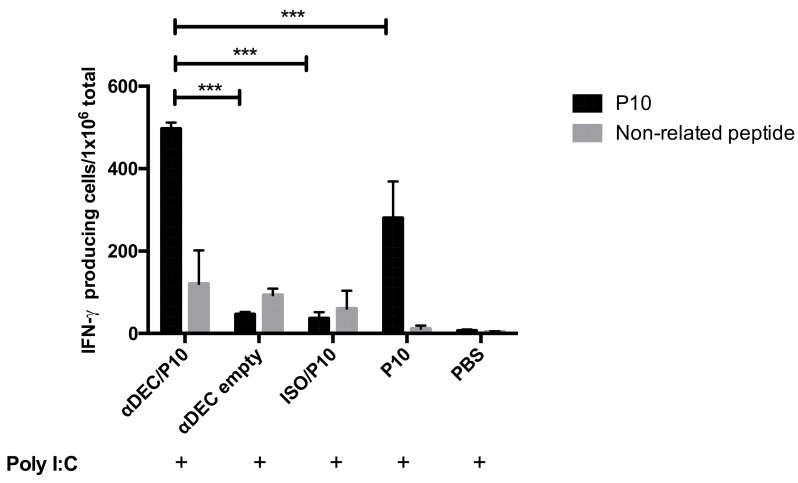
The production of IFN-γ by splenocytes from animals immunized significantly increases after challenge by αDEC/P10 monoclonal antibody. ELISPOT assay in which splenocytes from BALB/c mice 10 days after immunization with 5 μg of chimeric antibodies αDEC/P10 or αDECempty and ISO/P10 or P10 peptide in the presence of 50 mg of poly (I:C) or only with poly (I:C) were incubated in the presence of 20 μg/mL of P10 or an unrelated peptide (as negative control). The graph shows the number of cells producing IFN-γ per million splenocytes. The bars represent the means ± standard errors of one experiment performed in sextuplicate. *** Represents a significant difference of *p* < 0.001.

**Figure 3 jof-09-00548-f003:**
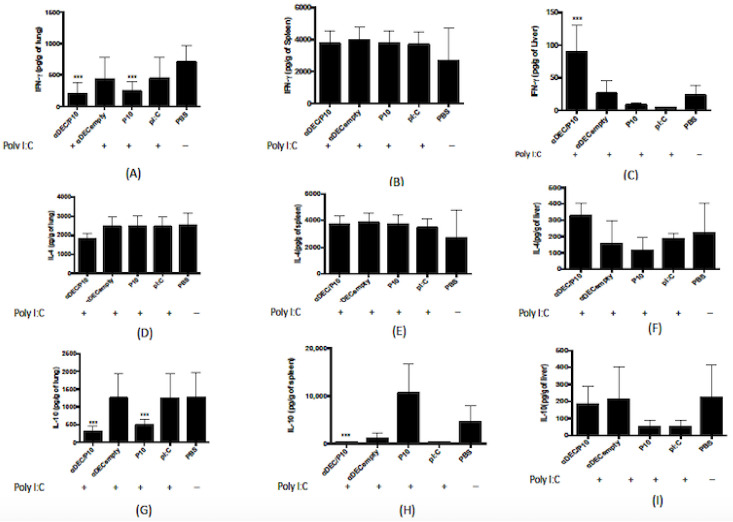
The cytokine production in the lungs, livers, and spleens of BALB/c mice infected with *P. brasiliensis* is significantly altered by the treatment with αDEC/P10 antibody. The cytokines IFN-γ (**A**,**D**,**G**), IL-4 (**B**,**E**,**H**), and IL-10 (**C**,**F**,**I**) were detected in homogenates of lung, spleen, and liver obtained from mice (*n* = 8) infected with 1 × 10^6^ yeasts of *P. brasiliensis* and treated with 3 sequential doses of the constructed chimeras (αDEC/P10 or αDECempty) given 7 days apart or P10 peptide in the presence of poly (I:C). Control groups included infected animals treated with poly (I:C) or PBS alone. All animals were analyzed 10 days after the 3rd injection of antibody or P10. Statistical analyses were performed via one-way ANOVA and as multiple comparisons via the Tukey test. *** *p* < 0.001, when compared with control (mice treated with PBS).

**Figure 4 jof-09-00548-f004:**
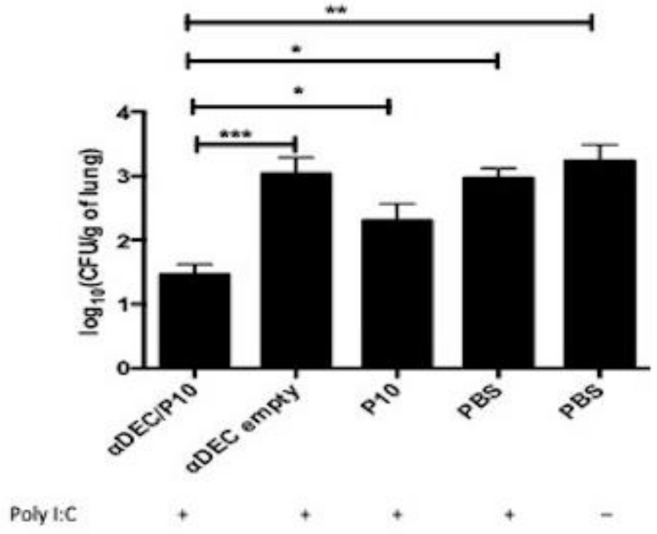
The DEC/P10 chimera reduces the fungal burden in the lungs of mice infected with *P. brasiliensis*. Mice were intratracheally infected with 1 × 10^6^
*P. brasiliensis* yeast cells and were treated 45 days later with 3 sequential doses of the constructed chimeras (αDEC/P10 or αDECempty) given 7 days apart or P10 peptide in the presence of poly (I:C) or their control (*n* = 8 mice/group). Ten days after treatment, the mice were euthanized. The results were expressed as log CFU/g (log of colony-forming units per gram of lung tissue). Statistical analyses were performed via one-way ANOVA, and as multiple comparisons via the Tukey test. * *p* < 0.05; ** *p* < 0.01; *** *p* < 0.001, when compared with control (mice treated with PBS).

**Figure 5 jof-09-00548-f005:**
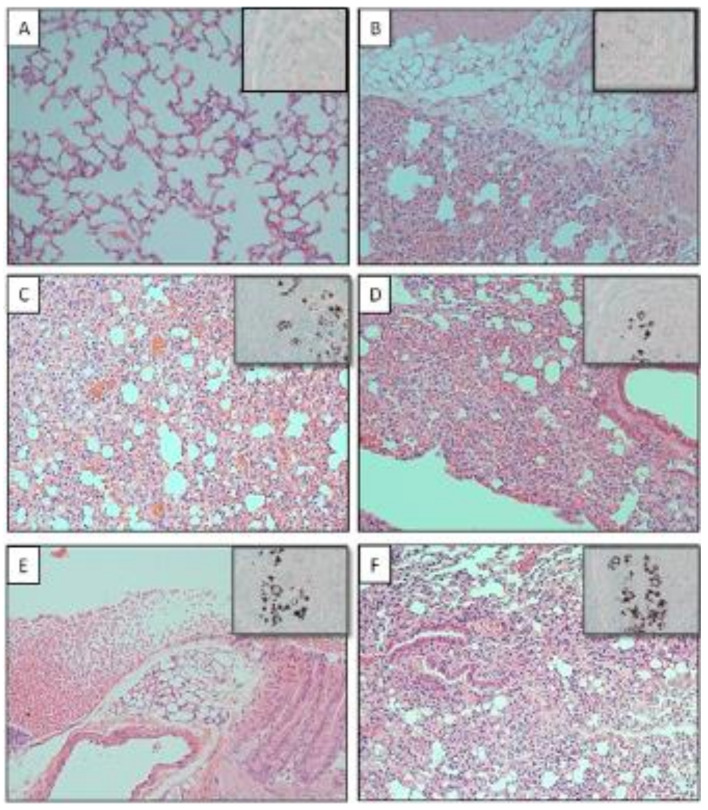
The DEC/P10 chimera reduces pulmonary fungal burden and preserves lung architecture in mice infected with *P. brasiliensis*. The experimental groups were intratracheally infected with 1 × 10^6^
*P. brasiliensis* yeast cells and were treated 45 days later with 3 sequential doses of the constructed chimeras (αDEC/P10, αDECempty, or ISO/P10) given 7 days apart or P10 peptide in the presence of poly (I:C) or their control (*n* = 8 mice/group). Ten days after treatment, the mice were euthanized. Histology of lung sections from infected BALB/c mice and treated: αDEC/P10 (**A**), αDECempty (**B**), isotype control (**C**), peptide P10 (**D**), poly (I:C) (**E**), and PBS (**F**). The tissues were stained with hematoxylin–eosin, and insets show Gomori silver staining (magnification was 400×).

## Data Availability

Data Availability Statements are available in section “MDPI Research Data Policies” at https://www.mdpi.com/ethics.

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
