# Peer review of "Targeting the P10 Peptide in Maturing Dendritic Cells via the DEC205 Receptor In Vivo: A New Therapeutic Strategy against Paracoccidioidomycosis"

_jof, 2023, doi:10.3390/jof9050548_

Round 1

Reviewer 1 Report

In the present study, the authors adapted a well-established technique to stimulate dendritic cells and drive Th1 responses. By combining a known CD4+ T cell epitope from Pb with an antibody that targets DEC205, the authors created a reagent that delivers immunodominant peptides to antigen presenting cells, as well as provides the necessary signals which allow dendritic cells to activate and differentiate CD4+ T cells. This reagent would serve as a promising vaccine against an important endemic mycosis in South America. The data and presentation of the data are overall solid. The following are a few concerns:

What is the purpose of gating on CD8- DC in Figure 1? What is represented as the “control” in this figure?

The labels on Figure 3 are too small to read. 3F&G have “***”, but it is not clear what this is in relation to.

Why does figure 2 (1 dose) and the rest of the figures (3 doses) use different vaccine schedules? It would make sense to provide ELISPOT data for splenocyte response after each of the  3 vaccinations to be consistent with the rest of the manuscript.

How are the cytokine responses in the spleen and liver relevant, if the lung is the focal point of infection? It is not clear why any relevant T cells would be producing cytokines in response to infection in remote organs.

There is significantly less fungal burden in the lungs of mice treated with aDEC/P10 (unequivocally, an exciting finding). However, this would explain the lackluster IFNg response reported in figure 3. It would be more meaningful to provide cytokine data across multiple time points, focusing exclusively on cytokines produced in the lungs. One would expect to see IFNg appear significantly higher and earlier in the aDEC/P10 mice than other groups, thereby augmenting protective immunity.

The vaccine was utilized therapeutically rather than prophylactically. Data on the prophylactic efficacy would be important, given that is goal of a majority of vaccine development.

Author Response

Reviewer # 1:

            First, we would like to thank you for your comments on our manuscript. Therefore, we are providing a point-by-point reply that indicates how we have revised the manuscript according to your comments:

1-“What is the purpose of gating on CD8- DC in Figure 1? What is represented as the “control” in this figure?”

- We used gating on CD8-DC because, among the several DC populations in the steady state, one expresses the C-type lectin endocytic receptor known as DEC205/CD205, the target of anti-DEC205-P10. The “control” on this figure is an anti-mouse IgG1; 

2-“The labels on Figure 3 are too small to read. 3F&G have “***,” but it is not clear what this is in relation to.”

  -We apologize for the labels in Fig 3. The information is “One-way ANOVA performed the statistical analysis, and as multiple comparisons by the Tukey test. * p<0.05; **p<0.01; ***<0.001

3-“Why does figure 2 (1 dose) and the rest of the figures (3 doses) use different vaccine schedules? It would make sense to provide ELISPOT data for splenocyte response after each of the  3 vaccinations to be consistent with the rest of the manuscript.”

-The experiments in figure 2 were performed only to evaluate the specific activation of the cellular response and IFN-gamma production and not with the purpose of vaccination as observed for the rest of the experiments; therefore, only one dose was used. 

4-“How are the cytokine responses in the spleen and liver relevant if the lung is the focal point of infection? It is not clear why any relevant T cells would be producing cytokines in response to infection in remote organs.”

-We agree with the comment that the lungs are essential, but we also believe that the lung would be vital in the early stages of the disease as it is the primary focus. However, in more extended periods and after the spread of the disease, other organs assume great importance, such as the spleen and liver; for this reason, we chose to analyze these organs. 

5-“There is significantly less fungal burden in the lungs of mice treated with aDEC/P10 (unequivocally, an exciting finding). However, this would explain the lackluster IFNg response reported in figure 3. It would be more meaningful to provide cytokine data across multiple time points, focusing exclusively on cytokines produced in the lungs. One would expect to see IFNg appear significantly higher and earlier in the aDEC/P10 mice than other groups, thereby augmenting protective immunity.”

-The low production of IFN-gamma associated with the decrease in the disease can be explained by the complete resolution of the lung lesions, as seen in fig 5. After the full resolution of the disease, we would expect a drop in the production of IFN-gamma, as observed in our results; 

6- “The vaccine was utilized therapeutically rather than prophylactically. Data on the prophylactic efficacy would be important, given that is goal of a majority of vaccine development.”

-With a better understanding of the epidemiology and pathophysiology of PCM, we understand that a prophylactic vaccine would be complex, as it would be imprecise to define the profile of the population that would be vaccinated. Therefore, in PCM, we believe that the therapeutic vaccine would be more useful when compared to the prophylactic one.

Reviewer 2 Report

Comments: 

This manuscript by Santos et al explains about a new chimeric antibody (αDEC/P10based therapeutic strategy against Paracoccidioidomycosis (PCM). The authors have targeted a fungal peptide P10 and fused it with a DEC205 receptor specific monoclonal antibody. The chimeric antibody was then used for immunization in PCM infected BALB/c mice. The chimeric antibody was able to bind with DCs which was evaluated by flow cytometry. A single immunization with chimeric antibody induced high levels of IFN gamma production in splenocytes, which was quantified by ELISPOT assay. Mice immunization also induced (IFN-gamma, IL-4, IL-10) T-cell dependent immune responses in liver, spleen and lungs. Mice pre-treated with chimeric antibody had low fungal burden in pulmonary tissue which was evaluated by CFU based and histopathological examinations. Overall, authors show that chimeric antibody based therapeutic strategy is a promising approach to combat PCM infection.

Specific Comments:

1.     Figure 1: What is the control used to show specific binding to CD8+ DCs? Please define.

2.     Figure 2: Additional control of αDECempty is missing.

3.     Figure 3: Resolution of graphs is too low to read or make out. Please improve.

4.     Figure 4: a. Mention p value between αDEC/P10 and P10 on graph, 

                b. Isotype control is absent

5.     Figure 5: Authors can include silver staining in 5A and 5B as well

Minor Comments

1. Authors have cited papers till 2017, latest information on Paracoccidioidomycosis can be included.

2. Line 22-23 is a duplicate of  Line 72-74, Please improve. 

3. Line 45-47, rewrite the sentence.

4. In Methodology section, use subscript and superscript for writing chemical names (Line 94).

5. Use symbol α in “aDEC/P10” (Line 100, Line 107).

6. In Figure 1, improve the resolution and alignment of different sections. Also use A,B,C,D… instead of flow chart.

7. Line 97: Include space between strain 18 and yeast cells.

8. Use full stop instead of commas, in P-values, “p<0,05; **p<0,01; ***<0,001” (Line 240 and Line 260).

9. Italicize “in vivo” and “in vitro” throughout the manuscript (Line 155, Line 290, Line 294).

10. In discussion section, authors can include current information regarding the status of vaccination approaches against PCM, and what are the advantages of proposed αDEC/P10 based therapeutics over previously developed/known therapeutics against PCM.

Author Response

Reviewer # 2:

            First, we would like to thank you for your comments on our manuscript. We are providing a point-by-point reply that indicates how we have revised the manuscript according to your comments:

Specific Comments:

  1. “ Figure 1: What is the control used to show specific binding to CD8+DCs? Please define. – “

-Was used an anti-mouse IgG1as control

  1. “Figure 2: Additional control of αDECempty is missing.”

-The additional control was added in the Fig 2;

  1. “ Figure 3: Resolution of graphs is too low to read or make out. Please improve.”

-The resolution of graphs was improved;

  1. “ Figure 4: a. Mention p value between αDEC/P10 and P10 on graph, “

-The value between αDEC/P10 and P10 was added

  1. “Figure 5: Authors can include silver staining in 5A and 5B as well –“

-The silver staining was added

Minor Comments

  1. “Authors have cited papers till 2017, latest information on Paracoccidioidomycosis can be included.”

 -The more recent information on PCM was added to the reviewed text

2.” Line 22-23 is a duplicate of  Line 72-74, Please improve. “

 -These sentences were rewrite;

3.” Line 45-47, rewrite the sentence.”

 -The sentence was rewrite;

  1. “In Methodology section, use subscript and superscript for writing chemical names (Line 94).”

 -This information was corrected;

  1. Use symbol α in “aDEC/P10” (Line 100, Line 107).”

 -This information was corrected;

  1. “In Figure 1, improve the resolution and alignment of different sections. Also use A,B,C,D… instead of flow chart.”

 -The resolution and alignment were improved;

  1. Line 97: Include space between strain 18 and yeast cells.”

 -This information was corrected;

  1. “Use full stop instead of commas, in P-values, “p<0,05; **p<0,01; ***<0,001” (Line 240 and Line 260).”

 -This information was corrected;

9.” Italicize “in vivo” and “in vitro” throughout the manuscript (Line 155, Line 290, Line 294).”

 -This information was corrected;

  1. “In discussion section, authors can include current information regarding the status of vaccination approaches against PCM, and what are the advantages of proposed αDEC/P10 based therapeutics over previously developed/known therapeutics against PCM.”

-As suggested, more recent information about vaccination against PCM was added.

Round 2

Reviewer 1 Report

The authors did not seem respond to many (most) of the comments with substantive changes to the manuscript. Please address the concerns raised in the manscript and clearly identify where these changes were made in the rebuttal letter.

Author Response

Answers to reviewers’ comments;

Reviewer # 1:

            First, we would like to thank you for your comments on our manuscript. Therefore, we are providing a point-by-point reply that indicates how we have revised the manuscript according to your comments:

1-“What is the purpose of gating on CD8- DC in Figure 1? What is represented as the “control” in this figure?”

- We used gating on CD8-DC because, among the several DC populations in the steady state, one expresses the C-type lectin endocytic receptor known as DEC205/CD205, the target of anti-DEC205-P10.; This information can be found between lines 107 and 111, page 3, in the Introduction section and has also been highlighted in the text to make it easier to find.

The “control” on this figure is an anti-mouse IgG1- This information can be found between lines 630 and 631, page 18, in the Legends of Figures section and has also been highlighted in the text to make it easier to find.

2-“The labels on Figure 3 are too small to read. 3F&G have “***,” but it is not clear what this is in relation to.”

  -We apologize for the labels in Fig 3, the figure was improved. This information has been improved and added in the Legends of Figures (Line 649, page 18) “One-way ANOVA performed the statistical analysis, and as multiple comparisons by the Tukey test. * p<0.05; **p<0.01; ***<0.001, when compared with mice treated with PBS (control).

3-“Why does figure 2 (1 dose) and the rest of the figures (3 doses) use different vaccine schedules? It would make sense to provide ELISPOT data for splenocyte response after each of the  3 vaccinations to be consistent with the rest of the manuscript.”

-The experiments in figure 2 were performed only to evaluate the specific activation of the cellular response and IFN-gamma production and not with the purpose of vaccination as observed for the rest of the experiments; therefore, only one dose was used. This information was added to the text. The statement “In order to analyze whether the constructed hybrid αDEC/P10 chimera could activate IFN-γ-producing cells in vivo, the animals were immunized and restimulated in vitro.” could be found between lines 253 and 254, page 7, in the Results section and has also been highlighted in the text to make it easier to find.

4-“How are the cytokine responses in the spleen and liver relevant if the lung is the focal point of infection? It is not clear why any relevant T cells would be producing cytokines in response to infection in remote organs.”

-We agree with the comment that the lungs are essential, but we also believe that the lung would be vital in the early stages of the disease as it is the primary focus. However, in more extended periods and after the spread of the disease, other organs assume great importance, such as the spleen and liver; for this reason, we chose to analyze these organs. This information has been improved and added in the Discussion section. The statement “The PCM can present as a severe and disseminated form involving the lungs, skin, lymph nodes, spleen, liver, and lymphoid organs of the gastrointestinal tract [34]. In order to analyze the immune response in the different organs such as the lung, spleen, and liver the cytokine production was quantifiedcould be found between lines 307 and 309, page 9 in the Discussion section and has also been highlighted in the text to make it easier to find;

5-“There is significantly less fungal burden in the lungs of mice treated with aDEC/P10 (unequivocally, an exciting finding). However, this would explain the lackluster IFNg response reported in figure 3. It would be more meaningful to provide cytokine data across multiple time points, focusing exclusively on cytokines produced in the lungs. One would expect to see IFNg appear significantly higher and earlier in the aDEC/P10 mice than other groups, thereby augmenting protective immunity.”

-The low production of IFN-gamma associated with the decrease in the disease can be explained by the complete resolution of the lung lesions, as seen in fig 5. After the full resolution of the disease, we would expect a drop in the production of IFN-gamma, as observed in our results; The information “We believe that this low amount of IL-10 and IFN-γ is because the disease is being controlled by the treatment. Assessing the levels of these cytokines at earlier times during the infection and treatment may reveal additional differences in the biological effects of the αDEC/P10 chimera.” could be found between lines 324 and 327, page 9 in the Discussion section;

6- “The vaccine was utilized therapeutically rather than prophylactically. Data on the prophylactic efficacy would be important, given that is goal of a majority of vaccine development.”

-With a better understanding of the epidemiology and pathophysiology of PCM, we understand that a prophylactic vaccine would be complex, as it would be imprecise to define the profile of the population that would be vaccinated. Therefore, in PCM, we believe that the therapeutic vaccine would be more useful when compared to the prophylactic one. The statement was added to the text “Chemotherapy is the basis of treatment of PCM in its various forms, however, can be extended for long periods with an alarming frequency of relapses. There is a consensus that the therapeutic vaccine instead of the prophylactic one would be the best way to control PCM. Several works have demonstrated that a therapeutic vaccine to boost the cellular immune response seemed relevant not only to reduce the time of treatment but also to prevent relapses and improve the prognosis of anergic cases” and could be found between lines 289 and 293, page 8 in the Discussion section

Sure that all comments were carefully considered and that now the manuscript fits the requirements made by the referees, and hoping to hear from you soon,

Sincerely Yours

Sandro Rogério Almeida PhD
